# Serum Procalcitonin Levels in Newly Diagnosed Hodgkin Lymphoma: Correlation with Other Inflammatory Biomarkers

**DOI:** 10.3390/medicina58101331

**Published:** 2022-09-22

**Authors:** Alexia Piperidou, Eleftherios Zografos, Theodoros J. Vassilakopoulos, Maria K. Angelopoulou, Georgios Hillas, Vassiliki Pappis, George Boutsikas, Maria Dimou, Kyriaki Petevi, Alexandros Kanellopoulos, Marie-Christine Kyrtsonis, Panayiotis Panayiotidis, Theodoros P. Vassilakopoulos

**Affiliations:** 1Department of Haematology and Bone Marrow Transplantation, School of Medicine, National and Kapodistrian University of Athens, Laikon General Hospital, 11527 Athens, Greece; 2Third Department of Critical Care Medicine, Evgenideio Hospital, 11528 Athens, Greece; 3Fifth Respiratory Medicine Department, Sotiria Chest Hospital, 11527 Athens, Greece; 4Haematology Clinical Trial Unit, First Department of Propaedeutic and Internal Medicine, Laikon General Hospital, National and Kapodistrian University of Athens, 11527 Athens, Greece

**Keywords:** procalcitonin, C-reactive protein, Hodgkin lymphoma, inflammation

## Abstract

*Background and Objectives*: Procalcitonin (PCT) is a useful biomarker for the diagnosis of sepsis. Inflammatory markers are elevated in patients with Hodgkin lymphoma (HL), and yet ongoing infection rarely coexists at diagnosis. PCT levels might be helpful in differentiating bacterial from disease-related inflammation. *Materials and Methods*: We evaluated serum PCT levels and other inflammation markers in newly diagnosed HL patients. Values < 0.50 ng/mL were considered normal (0.10–0.50 ng/mL: detectable, <0.10 ng/mL: undetectable), while values ≥ 0.50 ng/L were considered elevated. *Results:* Among 137 patients, 55 had B symptoms (40%), 77/130 (59%) had elevated Erythrocyte Sedimentation Rate (ESR) and 116 (85%) had elevated C-Reactive Protein (CRP) (median 38.1 mg/L (range; 2.97–328)). PCT levels were normal in most patients (undetectable 94/137 (68.5%) and detectable 41/137(30%)) with median value < 0.10 ng/mL (range; <0.10–15.90). Elevated PCT was recorded in only two patients (1.5%). Patients with PCT < 0.10 ng/mL had significantly lower median CRP (25.75; range (2.97–203.0)) compared to patients with PCT ≥ 0.1 ng/mL (median CRP 92.50 mg/L; range (3.34–328.0)). Almost all patients (40/41, 97.6%) with detectable PCT had elevated CRP. *Conclusions:* This is the first study showing that the inflammation characterizing HL is not associated with PCT elevations, although CRP levels are elevated in 85% of the patients. Normal PCT levels may rule out the possibility of occult infection, thus preventing extensive evaluation, which may delay treatment initiation.

## 1. Introduction

Procalcitonin (PCT) is the 116-amino prehormone of calcitonin that is normally synthesized and then secreted in bloodstream by the parafollicular C-cells of the thyroid in response to hypercalcemia. Under normal conditions, the produced PCT is almost entirely converted to its mature form, calcitonin, before entering the systemic circulation [1,2,3]. Thus, in healthy individuals, PCT is typically not detectable [3]. On the other hand, in systemic inflammation caused by bacterial infection, serum PCT levels are markedly elevated, through an alternative biologic pathway where non-thyroid tissue types, such as spleen, kidney, liver, pancreas, colon, brain and lungs, produce PCT [3,4,5]. Tissue cells are activated/triggered by cytokine-like stimulation factors (lipopolysaccharides, proinflammatory cytokines and endotoxins) and lack the ability to convert PCT to calcitonin [2,6]. As a result, PCT enters systemic circulation, causing a remarkable increase in serum levels [4,5,7].

In recent years, procalcitonin has emerged as a useful biomarker for the diagnosis of sepsis and bacterial infection [8]. Its specificity is superior to that of other common laboratory parameters such as leukocytosis, increased band cells and C-reactive protein (CRP), since serum PCT levels may increase within a few hours of inflammation and they typically reach their peak within 24 to 48 h [9,10]. In addition to bacterial infection and sepsis, increased serum PCT levels have been detected in patients with severe systemic fungal or parasitic infection, whereas viral infections are only moderately or not at all associated with PCT elevations. This is mainly attributed to cytokines produced in viral infections that block Tumor Necrosis Factor TNF-a production [9,11]. Consequently, serum PCT levels represent a useful and effective biomarker in the differentiation of bacterial from viral infections [1]. Moreover, elevated PCT may be correlated with the severity of multiple organ dysfunction syndrome, secondary to systemic inflammation and can be used in treatment monitoring and patients’ clinical follow-up [10,12].

In general, markers of underlying inflammation are elevated in the majority of patients with Hodgkin lymphoma (HL) being associated with disease activity and thus representing useful diagnostic and potential prognostic biomarkers [13,14,15,16]. More specifically, the laboratory evaluation of patients with HL at the time of diagnosis may reveal elevated Erythrocyte Sedimentation Rate (ESR ≥ 50 mm/h in 45% of patients and ESR ≥ 100 mm/h in 15% of patients), which is strongly associated with advanced stage and the presence of B symptoms [17,18,19]. There have also been described elevated CRP levels in about 70% of HL cases at the time of diagnosis, neutrophilic leukocytosis, as well as elevated serum ferritin, serum haptoglobin, plasma fibrinogen, serum Interleukin IL-6 or even serum hepcidin levels [13,19,20,21,22,23]. Often patients complain about fever or other constitutional symptoms, such as night sweats and/or weight loss—symptoms that are generally referred as B symptoms [24]. Thus, many physicians, especially those who are not familiar with HL, perform extensive laboratory work-up in order to rule out bacterial infections in these patients. However, an ongoing infection very rarely coexists with HL at the time of the disease diagnosis. The frequent elevation of inflammatory markers, especially CRP, which may be markedly increased, may pose diagnostic problems and unnecessary delay in treatment initiation. So far, there is limited published data evaluating serum PCT levels on previously untreated HL and their possible role in differentiating bacterial infection from non-bacterial inflammation. Based on the above, we investigated whether and to what extent the inflammatory condition underlying HL is associated with elevated serum PCT levels.

## 2. Materials and Methods

### 2.1. Patients and Staging

We conducted a retrospective cohort study evaluating patients (>14 years old) with newly diagnosed HL according to the revised 2008 World Health Organization (WHO) classification, who were referred to our department between April 2010 and August 2015. Patients were eligible for this study if they had previously untreated, biopsy-proven HL, had available serum PCT levels (routinely determined in most patients during the study period), were Human Immunodeficiency Virus (HIV)-negative and were free of any concurrent illness or any chronic inflammatory or active malignant disease. All patients were clinically assessed with detailed history and complete physical examination with emphasis on the presence of symptoms and signs of active infection. The Ann Arbor system was used for the clinical staging of the disease at the time of diagnosis [25]. Complete blood count, serum biochemical profile including serum protein electrophoresis, iron status, ESR, CRP, PCT, β_2_-microglobulin and hemolytic work-up were also obtained during the initial disease staging. Hemoglobin, white blood cell (WBC) count and biochemical markers were measured by standard clinical assays. Anemia was defined as hemoglobin levels < 11.5 g/dL in females and <13 g/dL in males. Furthermore, a unilateral bone marrow biopsy was performed on all patients. Imaging evaluation included chest X-ray and computed tomography of the chest and the abdomen as well as baseline Positron Emission Tomography (PET) scan. Further work-up for infectious diseases (blood/urine cultures) was only conducted if it was clinically indicated. No patient was treated with antibiotics without a documented infection.

Serum PCT levels were determined using chemiluminescence immunoassay (Liaison Brahms PCT; DiaSorin S.P.A., Saluggia, Italy). Values lower than 0.50 ng/mL were considered *normal*. Values between 0.10 and 0.50 ng/mL were considered *normal/detectable*, while values < 0.10 ng/mL were considered *normal*/*undetectable*. Serum PCT levels were considered *elevated* if they exceeded the cut-off value of 0.50 ng/L. For statistical analysis, we combined all patients with levels ≥ 0.10 ng/mL in a single group, since only two patients had PCT levels ≥ 0.50 ng/L. According to published data, the cut-off of 0.10 ng/mL, which falls within the normal range, may be clinically relevant, as values < 0.10 ng/mL (*normal/undetectable*) have been considered very unlikely to be associated with the likelihood of a bacterial infection. Procalcitonin values exceeding the cut-off burden of 0.5 ng/mL are associated with a very strong possibility of bacterial infection, while *normal/detectable* levels (values between 0.10 and 0.50 ng/mL) are considered unlikely (but not very unlikely) to be associated with bacterial infection. [26,27].

### 2.2. Statistical Analysis

Descriptive data analysis was performed on the following variables: B symptoms, Ann Arbor stage, histology type of HL, CRP, PCT, hemoglobin, WBC, lymphocyte and platelet count, ESR, serum albumin, Lactate dehydrogenase LDH, β_2_- microglobulin, serum ferritin, haptoglobulin, α_2_- and γ- globulin. The correlation between serum PCT levels, clinical and laboratory characteristics of the patients and other inflammatory markers was assessed using non-parametric methods (Mann–Whitney test and Spearman’s rho). A *p* < 0.05 was considered statistically significant. All statistical analyses were performed using IBM^®^ SPSS^®^ Statistics Edition 15.0.

## 3. Results

### 3.1. Patients’ Characteristics and Serum Inflammatory Marker Levels

The baseline characteristics of the study patients as well as laboratory findings at the time of the diagnosis are shown in Table 1. From April 2010 to August 2015, a total of 137 patients, who were diagnosed with HL and treated in our unit, were included in the study. The median age of patients was 36 years (15–88) and 69/137 (50%) were males. All patients had biopsy-proven HL and the most frequent histologic type was nodular sclerosis (NS) (96/137, 70%) following by the mixed cellularity (MC) subtype (30/137, 22%). Among 137 patients, 79 had advanced clinical stage (IB, IIB, III, IV; 58%) and 55 had B symptoms (40%) including fever, night sweats and weight loss.

Briefly concerning the laboratory findings, 116/137 (85%) patients had elevated CRP with a median value of 38.1 mg/L (2.97–328.0). The erythrocyte sedimentation rate was ≥50 mm/h in 77/130 (59%) patients. The median value for serum ferritin was 154.1 ng/mL (7, 6709), for serum haptoglobin was 298 mg/dL (66–774), and for α_2_- and γ-globulin levels were 0.96 g/L (0.59–1.51) and 1.47 g/L (0.69–3.75), respectively. Leukocytosis (WBC ≥ 15 × 10^9^/L) was recorded in 20/137 (15%) patients, severe lymphocytopenia (<0.6 × 10^9^/L) and hypoalbuminemia (<4 g/dL) in 16/136 (12%) and 104/137 (76%), respectively, while anemia was observed in 75/137 (55%) patients. Serum LDH and β_2_-microglobulin levels were elevated in 44/137 (32%) and 53/128 (41%) of the patients, respectively.

### 3.2. Serum PCT Levels

Serum PCT levels were normal in the vast majority of the patients (*normal/undetectable* 94/137(68.5%) and *normal/detectable* 41/137(30%)). The median value of the PCT levels in the study group was below 0.10 ng/mL (<0.10–15.90). *Elevated* PCT concentration (>0.50 ng/mL) was recorded in only 2 out of 137 (1.5%) patients. One severely ill patient with poor performance status had PCT levels of 15.30 ng/mL; however, this was associated with a concomitant confirmed staphylococcal bacteremia. The second patient had PCT levels of 1.60 ng/mL in the absence of documented infection. The patient was an 82-year- old male with advanced clinical stage (IIB) HL. Baseline laboratory tests revealed anemia (Hb 11.5 g/dL), leukocytosis (WBC 11.56 × 10^9^/L), just above-normal upper limit LDH levels (1.004-fold) and highly elevated inflammatory markers (ESR 109 mm/h, serum ferritin 164.50 ng/mL, haptoglobin 470 mg/dL, CRP 135 mg/L) probably associated with the severe inflammatory background of the underlying HL. All of the above markers were completely normalized after one week from chemotherapy initiation without any antibiotic treatment. The normalization of CRP levels without antibiotic therapy provides strong proof for the absence of an underlying infection. Excluding the patient suffering from a documented bacteremia, the serum PCT levels were *elevated* in only 1 out of 136 patients (0.7%) with HL in the absence of infection. In fact, the normal serum PCT levels ranged from *normal/undetectable* (<0.10 ng/mL) to levels just below the upper normal limit of 0.50 ng/mL (*normal/detectable*). A detailed description of the serum PCT levels is shown in Table 2.

### 3.3. Correlation between Serum PCT and CRP

Although the vast majority of newly diagnosed patients with HL and elevated CRP had normal PCT levels, there was a statistically significant association between these markers. The Spearman’s rho correlation coefficient was 0.471 (*p* < 0.001) and the corresponding scatter plot is shown Figure 1. Patients (96/137) who showed PCT values below 0.10 ng/mL had lower median CRP values (25.75, range (2.97–203.0)) compared to patients (41/137) with PCT levels above ≥0.10 ng/mL who had a median CRP value at 92.50 mg/L (3.34–328.0), and this association was statistically significant (*p* < 0.001). The correlation between serum PCT and CRP levels is presented in Table 2. The majority of patients (40/41, 97.6%) with detectable PCT levels also had elevated CRP (≥5 mg/L), whereas only 1 patient (2.4%) out of 21 with normal CRP showed detectable PCT levels.

### 3.4. Correlation between Serum PCT and Other Inflammatory and Non-Inflammatory Markers

The frequency of various markers characterizing HL within the two study groups according to the PCT levels (<0.10 ng/mL and ≥0.10 ng/mL) is shown in Table 2. Concerning the Ann Arbor stage, most patients (34/41, 83%) with PCT values ≥ 0.10 ng/mL were at an advanced disease stage compared to only 47% (45/96) in the *normal/undetectable* PCT group (*p* < 0.001). Moreover, B symptoms were present in the majority of HL patients with PCT ≥ 0.10 ng/mL (30/41, 73%) versus 25 out of 96 (26%) of patients with *normal/undetectable* PCT levels (*p* < 0.001). Compared to patients with *normal/undetectable* levels, those with PCT ≥ 0.10 ng/mL had more frequent ESR ≥ 50 mm/h (32/39, 82%), anemia (33/41, 81%), hypoalbuminemia (37/41, 90%), leukocytosis (11/41, 27%) and higher serum ferritin (median value: 361.4 ng/mL), haptoglobin (median value: 431.0 mg/dl) and a_2_-globulin (median value: 1.125 g/L) levels.

## 4. Discussion

PCT is the 116-amino acid prohormone of calcitonin, produced by the thyroid cells [1]. Both in vitro and in vivo studies have shown that not only bacterial endotoxins, but also various other mediators, such as TNF-α, IL-2, IL-6 and phytohemagglutinin, induce PCT production [3]. TNF-α and IL-6 are cytokines that rise during inflammation and are not specific to particular inflammation types. However, serum PCT rises specifically during bacterial inflammatory processes. Obviously, other underlying regulatory factors do exist, but still remain unknown [3]. Whether small or modest PCT elevations may also occur in non-bacterial inflammatory conditions such as HL is not known in the absence of published data.

Based on a meta-analysis, PCT has been found to be a more sensitive and specific biomarker compared to CRP in distinguishing bacterial from non-bacterial infections [28]. However, in another study, where PCT levels were measured in patients with community-acquired respiratory tract infections, serum PCT was not elevated [29]. Similar results were reported in a study where PCT was measured in children diagnosed with community-acquired pneumonia [30]. Both studies showed that PCT levels might not be as useful as a biomarker as CRP in less severe or localized infections. On the other hand, in patients with lower respiratory tract infections, PCT guided monitoring compared with standard follow-up guidelines resulted in lower rates of antibiotic exposure and antibiotic-associated adverse effects, indicating the diagnostic accuracy of PCT as a marker for suspected severe bacterial inflammation [31].

HL is frequently associated with cytokine-driven systemic inflammatory activity, characterized by fever or other B symptoms and the elevation of inflammatory markers, as mentioned above. Serum PCT levels have not been studied in HL so far. In the present study, the serum PCT levels were evaluated in 137 patients with newly diagnosed HL. The procalcitonin levels were normal (<0.5 ng/mL) in the overwhelming majority of patients (135 out of 137), irrespective of the severity of the inflammatory HL background, as reflected by CRP and other markers. Only one patient with documented staphylococcal bacteremia had PCT levels above 0.50 ng/mL. In another patient, the serum PCT levels were only mildly elevated, and were rapidly normalized following chemotherapy initiation without antibiotic treatment. Moreover, following therapy, inflammatory markers were normalized in all patients and B symptoms resolved.

Our findings suggest that the main pathway responsible for PCT elevation may not be involved in the HL inflammatory response, as reflected by the normal PCT levels in our study population. However, it should be noted that even though normal (<0.5 ng/mL) serum PCT levels were higher (and detectable) in patients with more severe inflammatory profile; patients with highly elevated CRP had serum PCT levels above 0.1 ng/mL much more frequently. Interestingly, PCT levels have been shown to be stimulated by proinflammatory mediators such as TNF-α and IL-6 as well. Both serum TNF-α and IL-6 are elevated in patients with previously untreated HL, especially in advanced-stage and symptomatic disease [20,23,32]. Thus, TNF-α and IL-6 may lead to limited PCT production in patients with highly active HL, which is not sufficient to cause clinically significant serum PCT elevation (≥0.5 ng/mL), except from “extreme” cases with advanced disease, presence of B symptoms and highly elevated inflammatory markers, as the single case observed in our series.

Although serum PCT levels have not been studied in HL so far, there is a single study assessing PCT in underlying hematological conditions associated with non-infectious inflammatory response. In patients with acute leukemia receiving high-dose cytarabine, drug-related fever is a common complication, which may be confused with bacterial or fungal infection, thus raising important therapeutic dilemmas. In a study of 57 children suffering from acute lymphoblastic leukemia or non-Hodgkin lymphoma (NHL) receiving high-dose cytarabine, PCT levels were analyzed in a randomly selected subgroup of 16 patients [33]. Blood samples for PCT evaluation were obtained at several time points before and after the start of Ara-C infusion. The median CRP level was 38 mg/L (3–150), while the serum PCT levels were detectable (≥0.1 ng/mL) in 15/16 and elevated (≥0.5 ng/mL) in 4 out of the 16 (25%) patients, all four of whom were febrile and had just been administered the fourth dose of Ara-C [33]. The pathogenesis of Ara-C fever is not fully known, but there are studies supporting that proinflammatory cytokines such as TNF-a, IL-6 and INF- γ may regulate this reaction [34]. As a result, the production of inflammatory markers such as CRP and PCT may be induced by these cytokines [35].

The data presented here may have crucial clinical implications. Normal serum PCT levels may rule out the diagnostic possibility of occult infection in patients with untreated HL with clinically significant CRP elevations and/or fever, thus preventing extensive investigation for the exclusion of underlying infection, which increases medical costs, the duration of hospitalization, patients’ anxiety and may further delay treatment initiation. According to these data, the measurement of PCT levels may also be very helpful as a biomarker in the setting of relapsed/refractory HL, where patients are frequently febrile and undergo aggressive chemotherapy with prolonged neutropenia, thus being exposed to an increased risk of infection.

## 5. Conclusions

In conclusion, this is the first study to support that the inflammatory condition underlying HL is not associated with meaningful serum PCT elevations, although CRP levels were elevated in 85% of them. Only extremely active disease may cause mild PCT elevations, but the frequency of this phenomenon does not exceed 1%. In contrast to CRP, serum PCT elevations should probably trigger a more extensive infection work-up.

## Figures and Tables

**Figure 1 medicina-58-01331-f001:**
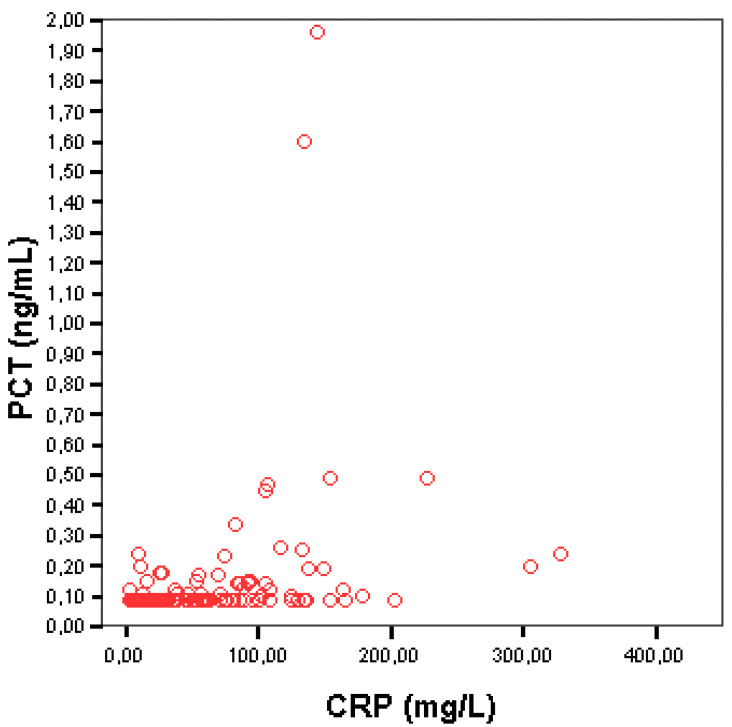
Correlation between serum PCT and CRP.

**Table 1 medicina-58-01331-t001:** Baseline characteristics of 137 patients with HL with available PCT levels.

Variable	All Patients (*n* = 137)
**Age**, years (median, IQR)	36 (15–88)
*<45 years*	89/137 (65%)
**Gender**	
*Male*	69/137 (50.4%)
**Histology**	
*Nodular sclerosis (NS)*	96/137 (70%)
*Mixed cellularity (MC)*	30/137 (22%)
*Lymphocyte rich (LR)*	3/137 (2%)
*Lymphocyte predominance (LP)*	1/137 (1%)
*Unclassified/Not otherwise specified (NOS)*	7/137 (5%)
**B symptoms**	
*Yes*	55/137 (40.1%)
**Ann Arbor stage**	
*I*	15/137 (11%)
*II*	62/137 (45%)
*III*	30/137 (22%)
*IV*	30/137 (22%)
*Early stage (IA/IIA)*	58/137 (42%)
*Advanced stage (IB/IIB/III/IV)*	79/137 (58%)
**Anemia**	
*Yes*	75/137 (55%)
**White blood cells (WBC)**	
*≥15 × 10^9^/L*	20/117 (15%)
**Lymphocytes**	
*<0.6 × 10^9^/L or <8%*	16/136 (12%)
**Platelet count** (median, IQR)	313 (31–705)
*≥400 × 10^9^/L*	35/137 (26%)
**ESR**	
*≥50 mm/h*	77/130 (59%)
**Serum albumin**	
*<4 g/dL*	104/137 (76%)
**LDH levels**	
*Elevated*	44/137 (32%)
**β_2_-microglobulin**	
*Elevated (>2.4 mg/L)*	53/128 (41%)
**CRP** (median, IQR)	38.1 (2.97–328.0)
*≥5 mg/L*	116/137 (85%)
**Serum Ferritin**, ng/mL (median, IQR *n* = 129)	154.1 (7–6709)
**Haptoglobin**, mg/dL (median, IQR *n* = 123)	298 (66–774)
**α_2_-globulin**, g/L (median, IQR *n* = 135)	0.96 (0.59–1.51)
**γ-globulin**, g/L (median, IQR *n* = 135)	1.47 (0.69–3.75)

WBC: white blood cells, CRP: C-reactive protein, IQR: interquartile range, ESR: erythrocyte sedimentation rate, LDH: lactate dehydrogenase.

**Table 2 medicina-58-01331-t002:** Demographic, clinical and laboratory findings, including inflammatory markers according to PCT concentration in patients with HL.

	PCT < 0.10 ng/mL	PCT ≥ 0.10 ng/mL *	*p*
**Total**, *n* (%)	96	41 *	
**Age**, years (median)	33.5	39	0.37
**Gender** (male (#, %))	42/96 (44%)	27/41 (66%)	0.018
**Ann Arbor stage** (advanced, IB/IIB/III/IV (#, %))	45/96 (47%)	34/41 (83%)	<0.001
**B symptoms** (yes (#, %))	25/96 (26%)	30/41 (73%)	<0.001
**Histology** (NS (#, %))	66/94 (70%)	30/39 (76%)	0.74
**Anemia**, g/dL (#, %)	42/96 (44%)	33/41 (81%)	<0.001
**WBC**, ×10^9^/L (≥15 × 10^9^ (#, %))	9/96 (9%)	11/41 (27%)	0.008
**Lymphocytes**, × 10^9^/L (<0.6 × 10^9^/L or <8%, (#, %))	7/96 (7%)	9/41 (22%)	0.014
**CRP** (median, range)	25.75 (2.97–203.0)	92.50 (3.34,328.0)	0.001 ^a^
*IQR*	7.96–57.80	50.25–130.5	
*CRP < 5 mg/L (%)*	20/96 (20.8%)	1/41 (2.4%)	0.006 ^b^
*CRP ≥ 5 mg/L (%)*	76/96 (79.2%)	40/41 (97.6%)	
*Highest CRP (mg/L)*	203	328	
**ESR**, mm/h (≥50 mm/h, (#, %))	45/91 (50%)	32/39 (82%)	0.001
**Albumin**, g/dl (*<4 g/dL (#, %))*	67/96 (70%)	37/41 (90%)	0.01
**β_2_-microglobulin**, mg/l (>2.4 mg/L, (#, %))	29/88 (33%)	24/40 (60%)	0.004
**LDH** (elevated (#, %))	31/96 (32%)	13/41 (32%)	0.95
**Platelet count**, x10^9^/L (median, range)	297.5 (96–662)	342.0 (31–705)	0.059
**Serum Ferritin**, ng/mL (median, range)	134.9 (7–1842)	361.4 (15.3–6709)	<0.001
**Haptoglobin**, mg/dL (median, range)	267.5 (66–617)	431.0 (119–774)	0.001
**α_2_- globulin**, g/L (median, range)	0.920 (0.60–1.50)	1.125 (0.59–1.51)	0.006
**γ- globulin**, g/L (median, range)	1.420 (0.69–375)	1.595 (0.79–3.47)	0.057

WBC: white blood cells, CRP: C-reactive protein, IQR: interquartile range, ESR: erythrocyte sedimentation rate, LDH: lactate dehydrogenase. * Two out of 41 patients (1.5% of the total 137 patients) had clearly abnormal PCT levels (≥0.50 ng/mL). **^a^** By Mann–Whitney test. **^b^** By chi-square test. The symbol # stands for the number entered before the percent sign.

## Data Availability

The data presented in this study are available on request from the corresponding author.

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
