# Peer review of "Serum Procalcitonin Levels in Newly Diagnosed Hodgkin Lymphoma: Correlation with Other Inflammatory Biomarkers"

_medicina, 2022, doi:10.3390/medicina58101331_

Round 1

Reviewer 1 Report

Dear Author (s)

1. It was not better that you selected a control group.

2. What is the reference for "PCT< 0.10; PCT ≥ 0.10".

3. Please match tables. Table 2 and Table 3 to Table 4.

4. Please change the site of the tables. Please write Table 4 before Tables 2 and 3.

5. Please calculate the area under the curve.

6. Please add standard deviation for the variables.

7. Please write "P" for "r-value" in italic and lower case.

8. Please add clinical significance.

9. Procalcitonin (PCT) is a biomarker that is released in response to bacterial infections. You could add infection status as a variable.

10. Why do you select CRP as a variable? What about other inflammatory factors?

Reviewer 2 Report

1.       Patients and Staging line 80  Why was this age range determined as the non-pediatric (>14 years) group? 16 or 18 is taken in many countries. Is there a specific reference?

2.       Line 180  correlation chart would be appropriate.

Reviewer 3 Report

A Piperidou et al performed an interesting study on the value of serum procalcitonin in patients newly diagnosed with Hodgkin Lymphoma. The authors demonstrated that this prehormone is not elevated in the vast majority of patients with HL. The study is well designed and the paper well written and easy to understand.

I have only a few minor comments.

-          More complete information of the statistical package should be given, including version and company.

-          Although the study seems to be done retrospectively, this is not clearly stated in patients and methods section. Please clarify it. Were HIV-infected patients included in the study?

-          In the section Results, the information shown in the text and in the tables, in some instances, is repeated. It is sufficient to show it once (in the case of Table 2, all the information is duplicated).

-          In Table 1 it is not necessary to show the complementary result in some variables such as age, gender, B-symptoms, anemia (e.g. if 50.4% of patients are male is taken for granted that 49,6% are women). It seems that the 3 last variables of the table are given as median (range). If so, this should be added. Please add serum LDH.

-          In Table 4, for the results given by median, ranges should be added.

-          I suggest to modify the title of Table 4: Inflammatory and non-inflammatory markers according PCT to concentration in patients with HL.

-          In Discussion, could the authors hypothesize why TNF-α and IL-6 are not increasing PCT in HL if these cytokines are supposed to activate its synthesis and have been demonstrated to be elevated in patients with HL?
